# Maternal Low-Protein Diet during Puberty and Adulthood Aggravates Lipid Metabolism of Their Offspring Fed a High-Fat Diet in Mice

**DOI:** 10.3390/nu14194057

**Published:** 2022-09-29

**Authors:** Xiaohua Huang, Yong Zhuo, Dandan Jiang, Yingguo Zhu, Zhengfeng Fang, Lianqiang Che, Yan Lin, Shengyu Xu, Lun Hua, Yuanfeng Zou, Chao Huang, Lixia Li, De Wu, Bin Feng

**Affiliations:** 1Animal Nutrition Institute, Sichuan Agricultural University, Chengdu 611130, China; 2Key Laboratory of Animal Disease-Resistant Nutrition of Ministry of Education, Sichuan Agricultural University, Chengdu 611130, China; 3College of Veterinary Medicine, Sichuan Agricultural University, Chengdu 611130, China

**Keywords:** maternal diet, offspring, low-protein diet, insulin sensitivity, hepatic steatosis, cholesterol

## Abstract

A maternal low-protein (LP) diet during gestation and/or lactation results in metabolic syndrome in their offspring. Here, we investigated the effect of maternal LP diet during puberty and adulthood on the metabolic homeostasis of glucose and lipids in offspring. Female mice were fed with normal-protein (NP) diet or a LP diet for 11 weeks. Male offspring were then fed with a high-fat diet (NP-HFD and LP-HFD groups) or standard chow diet (NP-Chow and LP-Chow groups) for 4 months. Results showed that maternal LP diet during puberty and adulthood did not alter the insulin sensitivity and hepatic lipid homeostasis of their offspring under chow diet, but aggravated insulin resistance, hepatic steatosis, and hypercholesterolemia of offspring in response to a post-weaning HFD. Accordingly, transcriptomics study with offspring’s liver indicated that several genes related to glucose and lipid metabolism, including lipoprotein lipase (*Lpl*), long-chain acyl-CoA synthetase 1 (*Acsl1*), Apoprotein A1 (*Apoa1*), major urinary protein 19 (*Mup19*), cholesterol 7α hydroxylase (*Cyp7a1*) and fibroblast growth factor 1 (*Fgf1*), were changed by maternal LP diet. Taken together, maternal LP diet during puberty and adulthood could disarrange the expression of metabolic genes in the liver of offspring and aggravate insulin resistance and hepatic steatosis in offspring fed a HFD.

## 1. Introduction

With alterations in eating patterns and sedentary habit, non-alcoholic fatty liver disease (NAFLD), type 2 diabetes mellitus (T2DM), obesity, and associated clinical features, such as hyperlipidemia and insulin resistance, are becoming more prevalent than ever [1,2,3,4]. Classically, the environment and genes are key factors associated with metabolic diseases [4]. Recently, epidemiological studies have reported that maternal nutrition status, including over- and under-nutrition, rendered offspring vulnerable to metabolic diseases in adult life [5,6]. However, the mechanisms underlying the susceptibility in offspring from maternal exposure to malnutrition are far from clear.

Dietary protein has extensive physiological and nutritional functions, including supplying amino acids for protein synthesis, and regulating food intake, lipid and glucose metabolism, immune functions, and bone metabolism [7]. Therefore, investigating the consequences of dietary protein levels on human health is particularly important. In many low-income countries, the average dietary protein intake is below international recommendations [8,9]. Human studies have shown that dietary protein levels in the mother during pregnancy influence the health of their offspring [10,11]. Studies on mice indicated that maternal protein restriction during pregnancy alleviated systemic insulin resistance in offspring in response to a post-weaning high-fat diet (HFD) [12,13]. However, another study suggested that maternal protein restriction during gestation and lactation combined with a post-weaning HFD impaired insulin sensitivity and increased serum insulin levels in their offspring [14].

Most studies on maternal low-protein (LP) diet and offspring health have focused on gestation, as this period is the “key window” period for fetal development [15], with maternal nutrition passed via the placenta to critical offspring organs, including the brain, liver, and adipose tissue [16]. A recent review article summarized that maternal nutrition levels before gestation altered offspring phenotypes by changing oocytes maturity and quality [17]. In a recent study, consuming a maternal LP diet one ovulatory cycle prior to mating led to cardiovascular abnormalities in offspring [18]. Furthermore, another study showed that a maternal LP diet before pregnancy increased insulin sensitivity and reduced glucose concentration in their male offspring [19]. Our previous study indicated that an LP diet during puberty and adulthood reduced primordial follicle activation in female mice [20]. However, it is still unclear whether a maternal LP diet during puberty and adulthood affects the glucose and lipids metabolism of their offspring.

In this study, F_0_ female mice were fed with LP diet from 5- to 16-week of age, followed by mating with normal male mice, and then were fed with a normal diet during pregnancy and lactation. The F_1_ male offspring were fed a normal diet or a HFD. The results showed that a maternal LP diet during puberty and adulthood exerted no effect on glucose and lipids metabolism in offspring fed normal diets, but aggravated the insulin resistance and hepatic steatosis in offspring fed a HFD.

## 2. Materials and Methods

### 2.1. Animal Care and Experimental Design

All animal procedures were reviewed and approved by the Animal Care and Use Committee of Sichuan Agricultural University. Four-week-old male and female mice (C57BL/6J) were obtained from Vital River laboratory Animal Technology Co. Ltd. (Beijing, China). Mice were kept in a pathogen-free room with 22 °C and 60% of stable temperature and humidity. After one-week adaptation, female mice were randomly grouped into two groups and were fed a low protein diet (LP, 8% Casein) or normal protein diet (NP, 20% Casein) until 16 weeks of age. Body weight and food intake were measured biweekly. These mice were then mated with normal male mice at 16-week of age. All mated female mice were fed a standard diet during gestation and lactation. F_1_ male pups were weaned at 3 weeks of age and then were accessed to a standard chow diet (10% calorie from fat, Research Diet; NP-Chow and LP-Chow groups) or a high-fat diet (60% calorie from fat, Research Diet; NP-HFD and LP-HFD groups) from 4 weeks of age (Figure 1A). Body weight was measured biweekly. Insulin tolerance test (ITT) was performed at 16 weeks of age. Blood glucose levels (Blood Glucose Strip, Yicheng, Beijing, China) were measured under fast condition at 20 weeks of age. Serum and tissues were then collected on the same day after blood glucose level measurement for further analysis.

### 2.2. Serum Biochemical Analysis

Serum levels of triglyceride (TAG) (Maccura biotechnology Co., Ltd., Chengdu, China), total cholesterol (TC) (Maccura), high-density lipoprotein cholesterol (HDL-C) (Kehua Bio-Engineering, Shanghai, China), and low-density lipoprotein cholesterol (LDL-C) (Kehua) were analyzed on an automatic biochemical analyzer (7020, HITACHI, Tokyo, Japan) with their respective kits [21]. Serum insulin level (ALPCO) and liver TAG content (Sigma, St. Louis, MO, USA) were analyzed with their respective commercial kits according to manufacturer’s instructions.

### 2.3. Liver Histological Analysis

Liver tissue was fixed in paraffin or OCT embedding media for H&E staining and Oil-red O staining, respectively. The samples were then cut into 4 μm pieces (RM2016, Leica, Shanghai, China). For H&E staining, dehydrated sections were stained with hematoxylin, rinsed with double-distilled water (ddH_2_O), then stained with eosin. The slices were then dehydrated and put on slides using a neutral resin. For Oil-red O staining, the slices were dried on slides, fixed in 4% paraformaldehyde for 15 min, rinsed in PBS, and stained with Oil-red O staining solution. After that, the slides were washed in PBS again, stained with hematoxylin, rinsed in ddH_2_O, dried, and mounted with glycerogelatin. Images were obtained with imaging software on a microscope (TS100, Nikon, Tokyo, Japan) with a CCD (DS-U3, Nikon) (NIS-Elements F3.2, Nikon).

### 2.4. Insulin Tolerance Test

Six-hour fasted mice were intraperitoneally injected with insulin (Novo Nordisk, Beijing, China) at the doses of 0.6 IU/kg (NP-Chow and LP-Chow groups) or 1.2 IU/kg (NP-HFD and LP-HFD groups) by intraperitoneal injection [22]. Blood glucose levels were measured at 0, 15, 30, 45, 60, and 90 min post insulin injection with a glucometer (Beijing Yicheng).

### 2.5. RNA Extraction and Real-Time PCR

RNA extraction and real-time PCR were performed as previously reported [23]. Briefly, liver powder was homogenized in TRizol reagent for RNA extraction (Invitrogen, Shanghai, China). The quality of RNA was analyzed by agarose gel and the concentration was measured with a spectrophotometer (NanoDrop 2000, Thermo Fisher Scientific, Waltham, MA, USA). 1 μg RNA was reverse-transcribed into cDNA with a reverse-transcription PCR kit according to the manufacturer’s instruction (Takara, Dalian, China). Real-time PCR was conducted on a quantitative-PCR machine (7500HT, ABI) with Power SYBR Green RT-PCR reagents (BioRad, Hercules, CA, USA). The conditions used for PCR were: 95 °C for 10 min for 1 cycle, and then 40 cycles of 95 °C for 15 s followed by 60 °C for 1 min. The real-time PCR data was analyzed by the 2^−ΔΔCt^ method with β-actin (*Actb*) as the reference. The sequences of the primers are listed in Appendix A.

### 2.6. Preparation of cDNA Libraries and Sequencing

cDNA libraries were constructed using the Illumina NEBNext UltraTM RNA Library Prep Kit. Briefly, four liver samples of each group from NP-HFD and LP-HFD groups were used. mRNA was extracted and purified from total RNA using Poly-T oligo-attached magnetic beads and was fragmented by heating in NEBNext First Strand Synthesis Reaction Buffer. First and second strand cDNAs were then synthetized from the mRNA. For hybridization, DNA fragments were linked with NEBNext adaptors after end-reparation and single nucleotide A (adenine) addition. After this, an AMPure XP system (Beckman Coulter, Beverly, CA, USA) was used to select cDNA fragments of 250 base pairs (bp)-300 bp. Libraries quantification and qualification were tested using the Agilent Bioanalyzer 2100 and ABI StepOnePlus Real-Time PCR System. The Illumina platform was then used to sequence the cDNA library.

### 2.7. Quality Control and Reads Mapping

To generate clean reads, low-quality reads in the raw data (more than 20% bases quality was lower than 10) were filtered, and then reads with adaptors and unknown bases (*n* bases more than 5%) were removed. Clean reads were mapped onto a reference genome (*mus musculus*) using HISAT2 (Hierarchical Indexing for Spliced Alignment of Transcripts, v2.0.5, UT Southwestern, Dallas, TX, USA).

### 2.8. Quantification and Analysis of Differentially Expressed Genes (DEGs)

Read numbers mapped to genes were counted using featureCounts (v1.5.0-p3). Then, the FPKM (fragments per kilobase of exon model per million mapped fragments) of each gene was calculated to reflect gene expression level.

DEGs between the two groups were analyzed using the DESeq2 R package (v1.16.1, European Molecular Biology Laboratory, Heidelberg, Germany). To control false discovery rates, *p*-values were adjusted using the Benjamini and Hochberg methods. Genes with corrected *p*-values less than 0.05 were considered as differentially expressed.

### 2.9. Gene Ontology (GO) and Kyoto Encyclopedia of Genes and Genomes (KEGG) Enrichment Analysis of DEGs

GO enrichment and KEGG pathways enrichment of the DEGs were analyzed using the clusterProfiler R package, and *p*-values of the GO terms less than 0.05 were considered significant.

### 2.10. Statistical Analysis

SAS 9.3 software was used to analyze the data (SAS Institute Inc., Cary, NC, USA). The data’s normality and homogeneity of variances were verified with a univariate test. An independent *t*-test was used to examine the differences between two groups with normal distribution data, whereas non-Gaussian and heterogeneous data were investigated using non-parametric analysis. The statistical difference of ITT was analyzed using repeated measures ANOVA. Results were presented as mean ± SE. *p*-value < 0.05 was considered significant.

## 3. Results

### 3.1. LP Diet Did Not Change the Food Intake and Body Weight in Female Mice

Female mice were fed with LP diet or NP diet from 5- to 16-week of age (Figure 1A). Results showed that the food intake was not changed by an LP diet, though the protein intake level was much lower in the LP group than that in the NP group (Appendix A). Moreover, the body weights were similar between the two groups (Appendix A).

### 3.2. Maternal LP Diet during Puberty and Adulthood Impaired Insulin Sensitivity in Offspring Fed High-Fat Diet

To investigate the effects of maternal protein restriction during puberty and adulthood on offspring metabolism, male F1 mice were fed a normal chow diet (NP-Chow and LP-Chow groups) or a high-fat diet (NP-HFD and LP-HFD groups) from 4 weeks old. Results showed that a maternal LP diet did not change the body weight and blood glucose levels of offspring under normal chow-diet or HFD (Figure 1B,C). However, systemic insulin sensitivity in offspring was impaired by the maternal LP diet under offspring HFD condition, whereas no difference was observed under normal chow-diet condition when compared with the maternal NP diet (Figure 1D,E). Moreover, the serum insulin level of the offspring from the LP group was higher than that of the offspring from the NP group under HFD conditions, whereas no difference was observed between the offspring from the LP and NP groups under normal chow-diet conditions (Figure 1F). Furthermore, the exogenous insulin-induced AKT phosphorylation level in the offspring’s liver was impaired by the maternal LP diet compared with the NP group under offspring HFD condition (Figure 1G,H). However, the expression of glucogenic genes *G6pc*, *Pepck1*, and *Pgc1α* in the liver were not changed by maternal protein restriction under neither normal chow-diet nor HFD (Figure 1I). These results indicated that the maternal LP diet potentially aggravated insulin resistance in HFD-fed offspring.

### 3.3. Maternal Low-Protein Diet during Puberty and Adulthood Aggravated Lipids Profiles in HFD-Fed Offspring

The homeostasis of lipids is important for animal health. The weight of liver and fat tissues in offspring were not changed by the maternal LP diet (Appendix A). However, liver triglycerides (TAG) content in offspring was increased by the maternal LP diet under HFD condition, as compared to the maternal NP diet (Figure 2A,B). Moreover, serum contents of TC, LDL-C, and HDL-C were higher, whereas TAG content was lower, in the offspring of the LP-HFD group than those of the NP-HFD group. However, no differences in serum lipids profiles were observed between the LP-Chow and NP-Chow groups (Figure 2C–F).

### 3.4. Transcriptome Profiling in Offspring’s Liver

To identify the potential mechanism involved in the regulation of offspring’s lipid metabolism by a maternal LP diet during puberty and adulthood, we performed transcriptome analysis to screen for differentially expressed genes (DEGs) in liver from HFD-fed offspring. In total, 24,800 genes were detected (Appendix A). With |log2 (fold changes)| > 0 and associated *p*-values < 0.05, a total of 388 DEGs were identified, including 200 up-regulated genes and 188 down-regulated genes, between the LP-HFD and NP-HFD groups (Figure 3A and Appendix A). Of these, 62 DEGs were very significantly different between the two groups (*p* < 0.01) (Appendix A). A heat map cluster analysis showed the distribution of DEGs (Figure 3B).

### 3.5. Functional and Pathway Enrichment Analysis of DEGs

GO functional and KEGG pathway enrichment analyses were performed with the DEGs between the LP-HFD and NP-HFD groups. Results showed that the DEGs were categorized into three groups (biological process (BP), cellular component (CC), and molecular function (MF)) based on GO enrichment analysis, and many of them were associated with lipids metabolism, such as long-chain fatty acid metabolic process, positive regulation of steroid metabolic process, regulation of lipid biosynthetic process, acetyl-CoA metabolic process, and cholesterol biosynthetic process in BP, lipid particle in CC, and steroid hydroxylase activity, cholesterol binding, long-chain fatty acid-CoA ligase activity, sterol transporter activity, lipid transporter activity, and lipoprotein particle receptor binding in MF (Figure 4A). Other high enriched GOs were positive regulation of vasculature development, cellular amino acid metabolic process, and bile acid biosynthetic process in BP, blood microparticle, melanosome, ruffle, cell leading edge, and pigment granule in CC, and carbohydrate binding, GTPase binding and carbohydrate transmembrane transporter activity in MF (Figure 4A and Appendix A).

KEGG pathway enrichment analysis showed that the major enriched pathways of the DEGs were metabolic pathways, including PPAR signaling pathway, steroid hormone biosynthesis, primary bile acid biosynthesis, cholesterol metabolism, fatty acid biosynthesis, and insulin resistance (Figure 4B). Other enriched pathways were complement and coagulation cascades, chemical carcinogenesis, prion diseases, protein processing in endoplasmic reticulum, cytokine-cytokine receptor interaction, and so on (Figure 4B and Appendix A).

### 3.6. Validation of the DEGs Linked to Lipid and Glucose Metabolism

Transcriptome analysis identified several DEGs that were previously linked to lipid and glucose metabolism (Table 1). These DEGs were interesting, as offspring mice in the LP-HFD group exhibited more liver fat accumulation and lower hepatic insulin sensitivity than the mice in the NP-HFD group (Figure 1G,H and Figure 2A,B). Therefore, we validated some of these DEGs using RT-qPCR. The selected genes were the regulator of glucose metabolism fibroblast growth factor 1 (*Fgf1*), the regulators of lipid and cholesterol metabolism including lipoprotein lipase (*Lpl*), long-chain acyl-CoA synthetase 1 (*Acsl1*), cholesterol 7α hydroxylase (*Cyp7a1*) and cell death inducing DFEA like effector A (*Cidea*), and an important regulator of both glucose and lipid metabolism major urinary protein 19 (*Mup19*). Interestingly, the expression of *Lpl*, *Cyp7a1*, and *Cidea* was upregulated, whereas the expression of *Mup19*, *Fgf1*, and *Acsl1* were downregulated in the liver of the LP-HFD group compared to those in the NP-HFD group (Figure 5). Additionally, the mRNA levels of other lipid metabolism genes were significantly upregulated by the maternal LP diet, including selectin P (*Selp*), StAR-related lipid transfer protein 6 (*Stard6*), and FMS-like tyrosine kinase 1 (*Flt1*) (Figure 5). The expression patterns of these genes were consistent with the result of the transcriptome data.

## 4. Discussion

Many studies have explored the effects of a maternal LP diet during gestation and/or lactation on the development and metabolism of their offspring. In this study, we investigated the effect of a maternal LP diet during puberty and adulthood, but not during gestation or lactation, on hepatic insulin sensitivity and lipids metabolism in their offspring. Results showed that offspring from the LP group exhibited hypercholesterolemia, impaired hepatic insulin sensitivity, increased serum insulin level, and liver fat accumulation, as compared to NP group, in response to HFD. In addition, our transcriptome analysis identified several DEGs that were previously reported to regulate lipid and glucose metabolism.

We observed that maternal LP consumption did not alter the body weight, or liver or white adipose tissue weight of offspring. A previous study also reported that maternal protein restriction before pregnancy did not change body mass, organ mass, glucose tolerance test, and glucose level in female offspring [19]. In another study with Wistar rats, maternal protein restriction (80 g protein/kg) for 8 weeks before pregnancy increased plasma glucose and TC levels in offspring under postnatal normal diet conditions [24]. Besides, when C57BL/6JBom female mice (8-week-old) were fed with a LP (8.4% protein) diet for 8 weeks, their male offspring had more fat deposit than the control mice under postnatal normal diet conditions [25]. The inconsistent results between different studies might be due to different animal models and/or feeding periods.

Liver is one of the main target tissues for insulin action. Hepatic insulin resistance decreases liver response to insulin signaling by interrupting the insulin-induced inhibition of hepatic glucose production, which leads to hyperlipidemia, hyperglycemia, liver disease, and type 2 diabetes [26,27]. Here, we observed that LP-HFD offspring exhibited lower insulin sensitivity and higher insulin levels when compared with NP-HFD offspring, even though body weight, liver and white adipose tissue weight, or blood glucose level were not significantly different between the two groups. The reason why the LP-HFD offspring had similar blood glucose levels compared to the NP-HFD offspring might be that the LP-HFD offspring exhibited reduced hepatic insulin sensitivity but higher serum insulin level. The reason why serum insulin levels increased in LP-HFD mice will be investigated in a future study. Moreover, we showed that a maternal LP diet during puberty and adulthood decreased *Fgf1* expression in the liver of post-weaning HFD-diet fed offspring. FGF1 is a hormone that functions as an insulin sensitizer. Previous studies reported that *Fgf1*-deficient mice developed insulin resistance under HFD conditions [28,29]. Critically, recombinant Fgf1 markedly reduced blood glucose levels in diabetic mice [30]. Therefore, a maternal LP diet during puberty and adulthood exerted a potential reprogramming effect on hepatic insulin sensitivity in the offspring by regulating the expression of *Fgf1*.

The liver is also an important hub for maintaining the balance of lipid metabolism. Lipids’ over-accumulation in the liver results in NAFLD pathogenesis [31]. We observed that the hepatic TAG content was higher, whereas the serum TAG level was lower in LP-HFD mice than those in the NP-HFD mice. This result suggested that the livers of the LP-HFD offspring might have had stronger lipids synthesis, lower lipids secretion, or higher TAG uptake than the livers of the HFD-NP offspring. Furthermore, RNA-sequencing data from the livers of LP-HFD and NP-HFD mice revealed several DEGs related to lipid synthesis and glucose metabolism, such as *Lpl*, *Acsl1*, *Cyp7a1*, and *Cidea*. The changes of several DEGs were confirmed by RT-qPCR. In particular, LP-HFD offspring had significantly reduced hepatic *Acsl1* mRNA level compared with NP-HFD offspring. ACSL1 is a major acyl-CoA synthetase in hepatic long-chain fatty acid processing and has a critical role in regulating fatty acid oxidation and uptake, triglyceride synthesis, and cholesterol metabolism [32,33]. Liver *Acsl1* deficiency induced severely hepatic triglyceride accumulation and hypercholesterolemia, particularly accompanied with hyperlipidemia [34]. Therefore, the change of *Acsl1* expression might be a reason for the accumulation of TAG in the liver of LP-HFD mice.

LPL is important for the clearance of blood TAG. In our study, the *Lpl* mRNA level was significantly increased in the liver of LP-HFD mice compared to the NP-HFD mice. This was consistent with the decreased level of TAG in the circulation of LP-HFD offspring. It was reported that *Lpl* gene mutations caused severe hypertriglyceridemia in humans [35], whereas liver *Lpl* deletion elevated plasma TAG level in mice [36]. Another study reported that *Lpl* overexpression in the liver exacerbated hepatic TAG accumulation and induced liver insulin resistance [37]. Thus, the decrease in blood TAG might due to the increased expression of hepatic *Lpl* in the offspring of LP diet fed dams.

Blood cholesterol level is important for health. Our study revealed that serum TC, LDL-C, and HDL-C levels were significantly increased in the LP-HFD offspring when compared with the NP-HFD offspring. Similarly, a previous study reported that a maternal LP diet during gestation resulted in hypercholesterolemia in offspring fed a high-fat and high-sucrose diet [38]. It was reported that CYP7a1 was involved in cholesterol metabolism by converting cholesterol into bile acids [39]. We observed that the expression of *Cyp7a1* was upregulated in the liver of LP-HFD offspring compared to the control mice. This might be due to the feedback of higher hepatic cholesterol level in LP-HFD offspring. CIDEa is a cell death-inducing DNA fragmentation factor alpha-like effector (CIDE) family member and was shown to be an important regulator of lipid storage and lipid droplet formation in hepatocytes. Previous studies reported that *Cidea* deficiency reduced TAG and TC levels in HFD-induced obese mouse and *ob*/*ob* genetic obese mouse liver via suppressing *de novo* lipid synthesis genes expression [40,41]. Furthermore, hepatic *Cidea* overexpression enhanced hepatic lipid accumulation [42]. When compared with our NP-HFD offspring, hepatic *Cidea* mRNA level was significantly increased in LP-HFD offspring. This could partly explain why the LP-HFD mice had higher lipid storage levels in the liver and serum cholesterol than the NP-HFD mice.

Mup19 has a critical role in glucose and lipid metabolism and is a liver-specific protein belonging to the Mup proteins family [43]. A previous study reported that Mup19 was putatively associated with lipid deposition in the liver [44], similar to Mup1, which reduced fat accumulation and insulin resistance in *db*/*db* mouse liver [45]. In our study, *Mup19* expression was significantly down-regulated in the liver of LP-HFD offspring when compared with the NP-HFD group. Thus, MUP19 might also play a role in the regulation of offspring’s glucose and lipids metabolism in LP-HFD offspring. Further studies are required to explore the epigenetic mechanisms of how maternal protein restriction during puberty and adulthood affects offspring metabolism. In particular, the common transcription factors of the *Fgf1*, *Acsl1*, *Lpl*, *Cidea*, and *Mup19* genes will be the main targets.

## 5. Conclusions

A maternal low-protein diet during puberty and adulthood could aggravate hepatic steatosis and insulin resistance in HFD-fed offspring, and this might be mediated by the alteration of metabolism-related genes. This study is of interest and shows that a maternal low-protein diet during puberty and adulthood is harmful to the health of their offspring.

## Figures and Tables

**Figure 1 nutrients-14-04057-f001:**
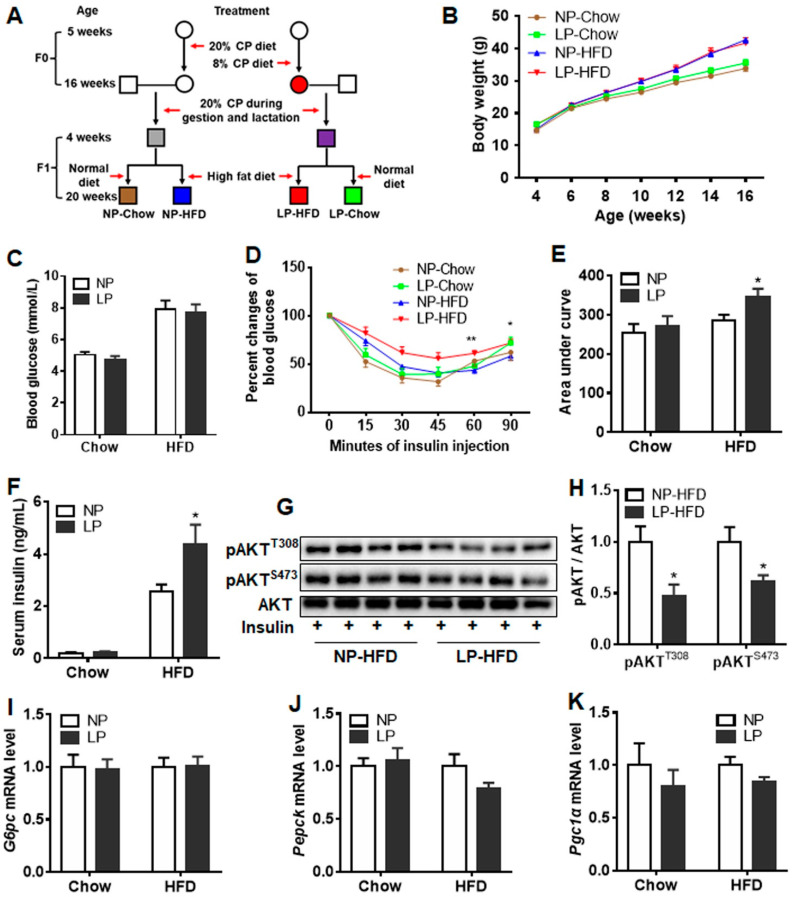
Maternal low-protein diet during puberty and adulthood impaired insulin sensitivity of offspring under HFD condition. C57/BL6J female mice were fed with NP diet or a LP diet for 11 weeks until 16 weeks of age. These mice were then mated with normal-diet fed male mice. All female mice were fed with normal-diet during pregnancy and lactation. F1 male pups were weaned at 3 weeks of age and then were accessed to normal chow diet or high-fat diet from 4 weeks of age. (**A**) Breeding strategy to generate the first-generation offspring. (**B**) Body weight of the offspring (*n* = 10–14 for each group). (**C**) Blood glucose levels at harvest (*n* = 7–9 for each group). (**D**) Insulin tolerance test and (**E**) the area under curve (*n* = 7 for each group). (**F**) Insulin levels in the serum (*n* = 6 for each group). (**G**) Western blot bands of phosphorylation AKT in liver tissue. (**H**) The quantification of phosphorylation AKT levels. The mRNA levels of *G6pc* (**I**), *Pepck* (**J**), and *Pgc1α* (**K**) in the liver of offspring (*n* = 7–9 for each group). NP, normal protein diet; LP, low protein diet. Data were expressed as mean ± SE. * *p* < 0.05, ** *p* < 0.01 LP vs. NP.

**Figure 2 nutrients-14-04057-f002:**
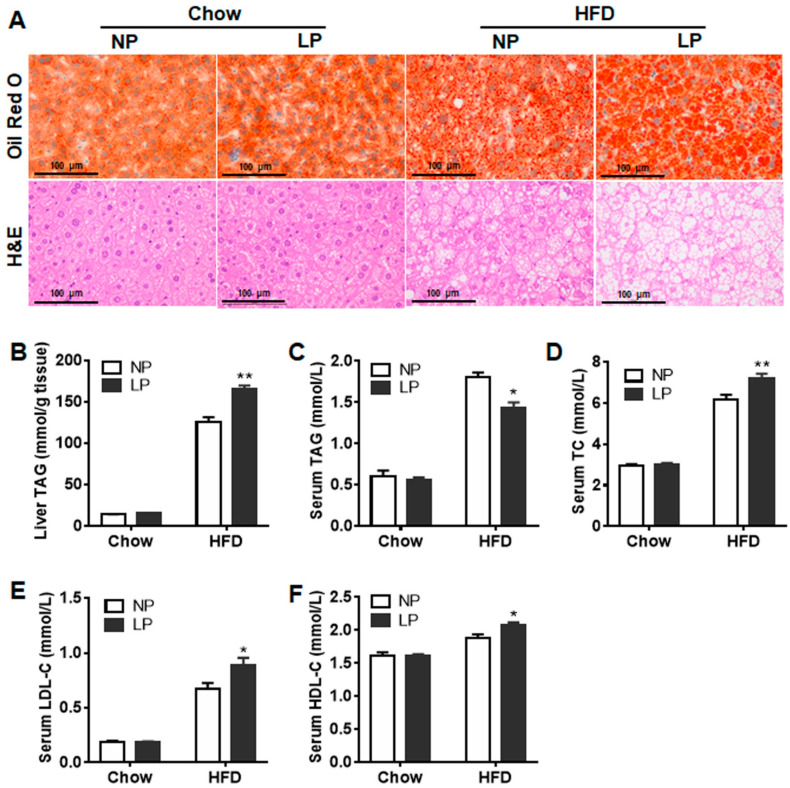
Maternal low-protein diet during puberty and adulthood aggravated hepatic steatosis of offspring under HFD condition. (**A**) Oil-red O staining and H&E staining with the liver of offspring. Bars equal to 100 μm. (**B**) TAG content in the liver. (**C**) TAG content in the serum. (**D**) TC content in the serum. (**E**) LDL-C content in the serum. (**F**) HDL-C content in the serum. *n* = 7–9 for each group. Data were expressed as mean ± SE. NP, normal protein diet; LP, low protein diet. * *p* < 0.05, ** *p* < 0.01 LP vs. NP.

**Figure 3 nutrients-14-04057-f003:**
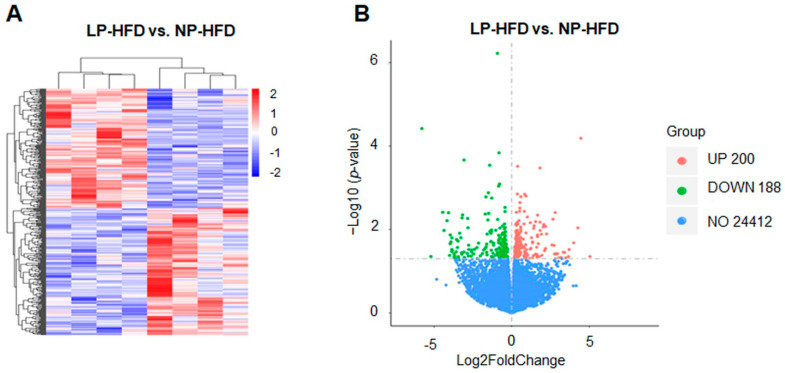
Analysis of DEGs in the liver between LP-HFD and NP-HFD groups. Transcriptome analysis was performed with the liver of LP-HFD and NP-HFD offspring. RNA-seq libraries were prepared with total RNAs extracted from liver tissues, and sequencing was carried out on Illumina Hiseq platform. The final data were analyzed by the DEseq2 R software. (**A**) Heatmap clustering was analyzed with differentially expressed genes between the LP-HFD and NP-HFD groups. (**B**) The threshold for differential expression genes (cut-off = |log2 Fold Change| > 0 and *p*-value < 0.05) is indicated by dashed black lines. Solid red and green dots represent the up-regulated and down-regulated genes, respectively. *n* = 4 for each group.

**Figure 4 nutrients-14-04057-f004:**
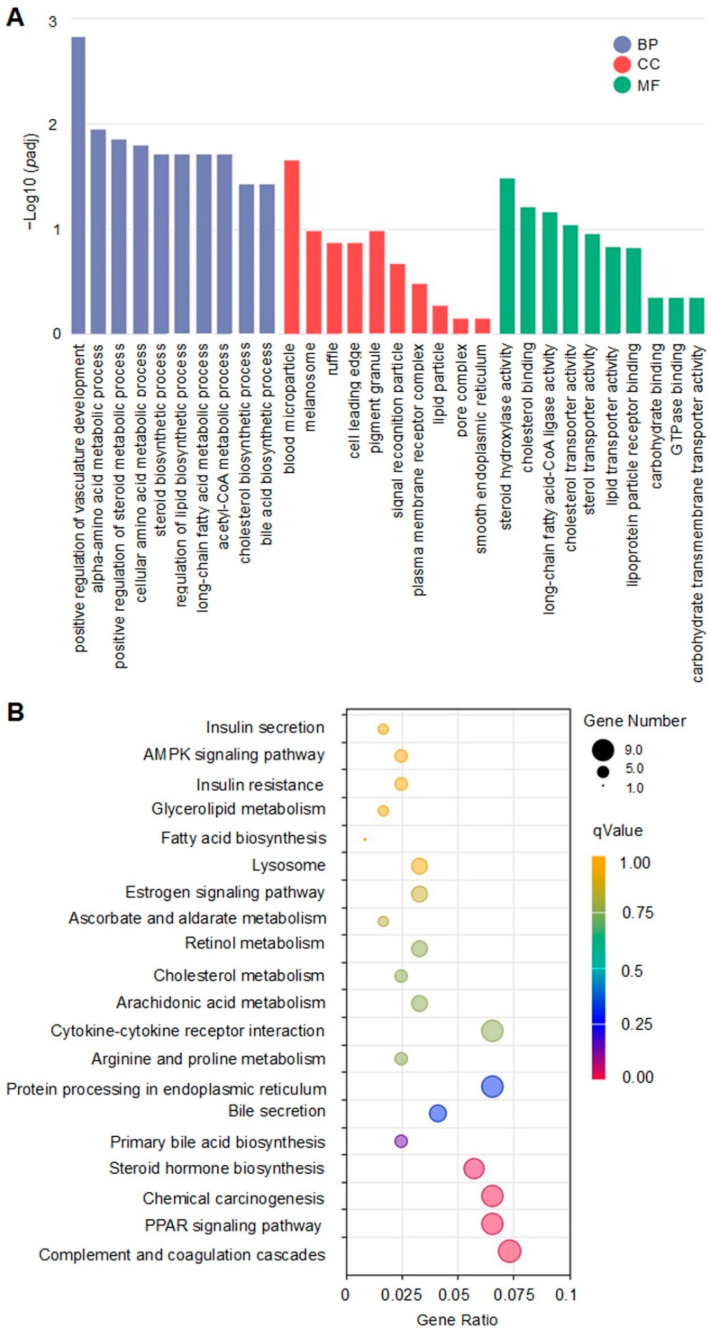
GO and KEGG pathway enrichment analysis of DEGs. (**A**) GO annotation and classification based on biological process (BP), cellular component (CC), and molecular function (MF). (**B**) KEGG enrichment bubble chart. The color of the point indicates the q-value of the hypergeometric test, and the size represents their corresponding number of genes in their pathway.

**Figure 5 nutrients-14-04057-f005:**
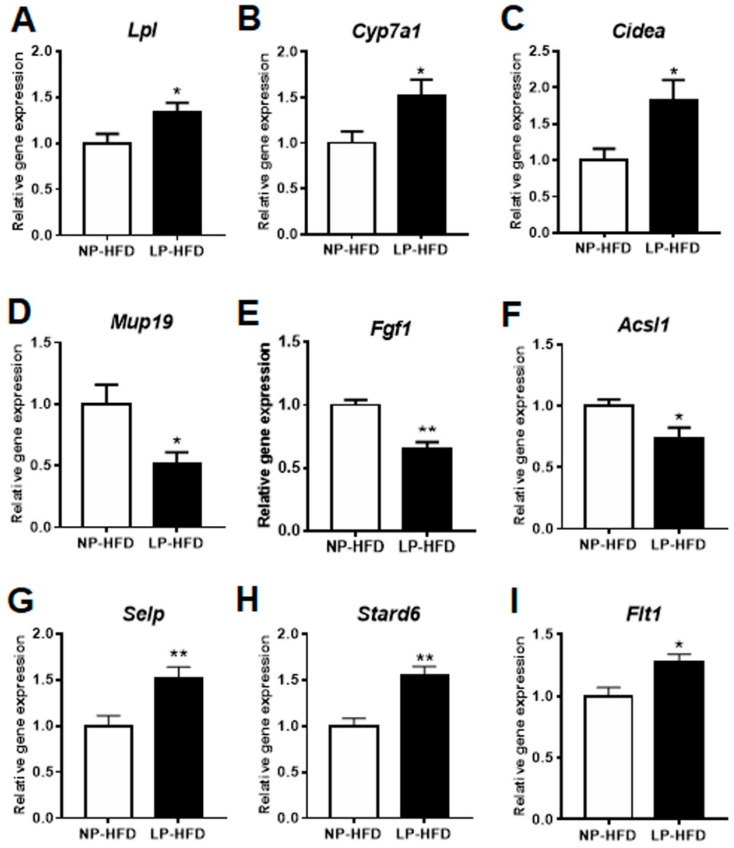
Validation of DEGs involved in lipid and glucose metabolism in the liver of NP-HFD and LP-HFD offspring. RT-qPCR was used to validate the RNA-seq data. The relative expression of *Lpl* (**A**), *Cyp7a1* (**B**), *Cidea* (**C**), *Mup19* (**D**), *Fgf1* (**E**), *Acsl1* (**F**), *Selp* (**G**), *Stard6* (**H**), and *Flt1* (**I**). *n* = 7–9 for each group. Data were expressed as mean ± SE. * *p* < 0.05; ** *p* < 0.01 LP-HFD vs. NP-HFD.

**Table 1 nutrients-14-04057-t001:** DEGs related to lipid and glucose metabolism.

Ensembl ID	Gene	Description	Log_2_FC	*p* Value
ENSMUSG00000078673	*Mup19*	major urinary protein 19	−4.46	0.0001
ENSMUSG00000036585	*Fgf1*	fibroblast growth factor 1	−0.35	0.0068
ENSMUSG00000028240	*Cyp7a1*	cytochrome P450, family 7, subfamily a, polypeptide 1	0.55	0.0131
ENSMUSG00000024378	*Stard4*	StAR-related lipid transfer (START) domain containing 4	−0.32	0.0157
ENSMUSG00000028195	*Cyr61*	cysteine rich protein 61	0.36	0.0265
ENSMUSG00000032083	*Apoa1*	apolipoprotein A-I	−0.24	0.0373
ENSMUSG00000005514	*Por*	P450 (cytochrome) oxidoreductase	−0.28	0.0426
ENSMUSG00000095320	*Ccl21a*	chemokine (C-C motif) ligand 21A (serine)	1.53	0.0059
ENSMUSG00000029648	*Flt1*	FMS-like tyrosine kinase 1	−0.30	0.0295
ENSMUSG00000024526	*Cidea*	cell death-inducing DNA fragmentation factor, alpha subunit-like effector A	0.91	0.000
ENSMUSG00000020644	*Id2*	inhibitor of DNA binding 2	−0.38	0.0305
ENSMUSG00000015568	*Lpl*	lipoprotein lipase	0.47	0.0201
ENSMUSG00000031767	*Nudt7*	nudix (nucleoside diphosphate linked moiety X)-type motif 7	−0.53	0.0119
ENSMUSG00000006517	*Mvd*	mevalonate (diphospho) decarboxylase	−0.46	0.0135
ENSMUSG00000057228	*Aadat*	aminoadipate aminotransferase	−0.29	0.0359
ENSMUSG00000028011	*Tdo2*	tryptophan 2,3-dioxygenase	−0.24	0.0407
ENSMUSG00000034593	*Myo5a*	myosin VA	0.76	0.0092
ENSMUSG00000043461	*Sptssb*	serine palmitoyltransferase, small subunit B	3.34	0.0122
ENSMUSG00000032231	*Anxa2*	annexin A2	−0.41	0.0134
ENSMUSG00000021135	*Slc10a1*	solute carrier family 10 (sodium/bile acid cotransporter family), member 1	−0.25	0.0162
ENSMUSG00000030382	*Slc27a5*	solute carrier family 27 (fatty acid transporter), member 5	−0.21	0.0252
ENSMUSG00000041828	*Abca8a*	ATP-binding cassette, sub-family A (ABC1), member 8a	−0.34	0.0343
ENSMUSG00000018796	*Acsl1*	acyl-CoA synthetase long-chain family member 1	−0.21	0.0366
ENSMUSG00000079608	*Stard6*	StAR-related lipid transfer (START) domain containing 6	−2.86	0.0378
ENSMUSG00000020484	*Xbp1*	X-box binding protein 1	−0.41	0.0120
ENSMUSG00000024292	*Cyp4f14*	cytochrome P450, family 4, subfamily f, polypeptide 14	−0.30	0.0039
ENSMUSG00000042248	*Cyp2c37*	cytochrome P450, family 2. Subfamily c, polypeptide 37	−0.41	0.0073
ENSMUSG00000092008	*Cyp2c69*	cytochrome P450, family 2, subfamily c, polypeptide 69	−2.07	0.0075
ENSMUSG00000019768	*Esr1*	estrogen receptor 1 (alpha)	−0.48	0.0276
ENSMUSG00000054827	*Cyp2c50*	cytochrome P450, family 2, subfamily c, polypeptide 50	−0.41	0.0179
ENSMUSG00000020258	*Glyctk*	glycerate kinase	−0.20	0.0437
ENSMUSG00000002831	*Plin4*	perilipin 4	0.42	0.0030
ENSMUSG00000028427	*Aqp7*	aquaporin 7	1.16	0.0238
ENSMUSG00000025006	*Sorbs1*	sorbin and SH3 domain containing 1	0.27	0.0285
ENSMUSG00000016194	*Hsd11b1*	hydroxysteroid 11-beta dehydrogenase 1	−0.28	0.0073
ENSMUSG00000035780	*Ugt2a3*	UDP glucuronosyltransferase 2 family, polypeptide A3	−0.31	0.0139
ENSMUSG00000039648	*Kyat1*	kynurenine aminotransferase 1	−0.29	0.0244
ENSMUSG00000030711	*Sult1a1*	sulfotransferase family 1A, phenol-preferring, member 1	−0.30	0.0262
ENSMUSG00000057425	*Ugt2b37*	UDP glucuronosyltransferase 2 family, polypeptide B37	−0.57	0.0362
ENSMUSG00000039519	*Cyp7b1*	cytochrome P450, family 7, subfamily b, polypeptide 1	−0.80	0.0422
ENSMUSG00000067144	*Slc22a7*	solute carrier family 22 (organic anion transporter), member 7	−0.91	0.0230
ENSMUSG00000041698	*Slco1a1*	solute carrier organic anion transporter family, member 1a1	−1.64	0.0419
ENSMUSG00000027938	*Creb3l4*	cAMP responsive element binding protein 3-like 4	1.58	0.0128
ENSMUSG00000038648	*Creb3l2*	cAMP responsive element binding protein 3-like 2	−0.36	0.0444
ENSMUSG00000078683	*Mup1*	major urinary protein 1	−1.96	0.0414
ENSMUSG00000078687	*Mup8*	major urinary protein 8	−2.80	0.0039
ENSMUSG00000096674	*Mup15*	major urinary protein 15	−2.03	0.0087
ENSMUSG00000082644	*Mup-ps19*	major urinary protein, pseudogene 19	−1.24	0.0280
ENSMUSG00000020053	*Igf1*	insulin-like growth factor 1	−0.29	0.0461
ENSMUSG00000024378	*Stard4*	StAR-related lipid transfer (START) domain containing 4	−0.32	0.0157

## Data Availability

The data presented in this study are available on request from the corresponding author.

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
