# Peer review of "Maternal Low-Protein Diet during Puberty and Adulthood Aggravates Lipid Metabolism of Their Offspring Fed a High-Fat Diet in Mice"

_nutrients, 2022, doi:10.3390/nu14194057_

Round 1

Reviewer 1 Report

The manuscript explored the effect of a low-protein maternal diet since puberty and adulthood in the metabolic and genic parameters in the young adult offspring, when they were challenged with a high fat diet. The research is innovative in terms of exploring the effect of a restricted protein diet not during the pregnancy or lactation (as it was broadly aborded) but an early exposition during puberty and adulthood.  

The researchers described that prior exposition with low protein diet imprints a deleterious gene fingerprint in the challenged high-fat diet offspring, which results in lower insulin sensitivity, higher insulin serum levels, diminished downstream phospho- Akt activity, higher TAG liver content, and higher serum levels of total cholesterol, LDL-C and HDL-C. Therefore, al, these alterations indicated. These metabolic alterations were also accompanied by a dysregulated profile of 388 liver genes in the areas of biological process, cellular components and molecular function.

The experimental design was well developed and clearly described. The results were also clearly presented.

I only have minor observations:

-           In Figure 1A, I suggest to label the four experimental groups that are managed throughout the manuscript (i.e. NP-Chow, NP-HFD, LP-Chow and LP-HFD). Accordingly, X labels should be NP-HFD or LP-HFD in all the panels of Figure 5.  

-           In Figure 2A, Magnification bars indicated 100 um but the Figure legend depicted that bars are equal to 50 um. Please could you clarify this point.

Author Response

The manuscript explored the effect of a low-protein maternal diet since puberty and adulthood in the metabolic and genic parameters in the young adult offspring, when they were challenged with a high fat diet. The research is innovative in terms of exploring the effect of a restricted protein diet not during the pregnancy or lactation (as it was broadly aborded) but an early exposition during puberty and adulthood.  

The researchers described that prior exposition with low protein diet imprints a deleterious gene fingerprint in the challenged high-fat diet offspring, which results in lower insulin sensitivity, higher insulin serum levels, diminished downstream phospho- Akt activity, higher TAG liver content, and higher serum levels of total cholesterol, LDL-C and HDL-C. Therefore, al, these alterations indicated. These metabolic alterations were also accompanied by a dysregulated profile of 388 liver genes in the areas of biological process, cellular components and molecular function.

The experimental design was well developed and clearly described. The results were also clearly presented.

Response: We thank this reviewer for the valuable comments.

I only have minor observations:

-           In Figure 1A, I suggest to label the four experimental groups that are managed throughout the manuscript (i.e. NP-Chow, NP-HFD, LP-Chow and LP-HFD). Accordingly, X labels should be NP-HFD or LP-HFD in all the panels of Figure 5. 

Response: Thanks for the valuable suggestion. The labels in Figure 1A and Figure 5 have been updated according to the suggestion of this reviewer.

-           In Figure 2A, Magnification bars indicated 100 um but the Figure legend depicted that bars are equal to 50 um. Please could you clarify this point.

Response: Thanks for the valuable comments. The bars equal to 100 μm. The figure legend has been updated.

Reviewer 2 Report

The submitted manuscript is about a topic that had been extensively studied, in the literature, there have been endless article with the effect of sub-optimal maternal diet on the fetal development. However, among them only few covers the adulthood risk of metabolic diseases. The manuscript by Feng and co-workers focused on a long term effect of maternal low-protein diet on the lipid metabolism during puberty and adulthood.
The Authors used a large variety of methodological approaches and the methods are sound, the analysed parameters are properly selected. The results are transparent and convincing. There lies only one question in my mind in the context of evaluating the figures. What is the reason behind the different sample number in case of the selected parameters (Figure 1; (B) N=10-14 for each group; (C) N=6-10 for each group; (D) and (E) N=7 for each group; (F) N=6 for each group).
I feel a bit of lack in the discussion and the conclusion sections, which is probably due to the presentation of the huge amount of data. A good number of individual parameters are mentioned in the discussion, but no broader connections were presented.

Minor:

there are some typing and/or grammar mistake in the text.

Figure legend 2: “Oil-red O staining and H&E staining Bars equal to 50 μm”, on the pictures bars are mark as 100 μm.

Author Response

The submitted manuscript is about a topic that had been extensively studied, in the literature, there have been endless article with the effect of sub-optimal maternal diet on the fetal development. However, among them only few covers the adulthood risk of metabolic diseases. The manuscript by Feng and co-workers focused on a long term effect of maternal low-protein diet on the lipid metabolism during puberty and adulthood.

Response: We thank this reviewer for the valuable comments.

The Authors used a large variety of methodological approaches and the methods are sound, the analysed parameters are properly selected. The results are transparent and convincing. There lies only one question in my mind in the context of evaluating the figures. What is the reason behind the different sample number in case of the selected parameters (Figure 1; (B) N=10-14 for each group; (C) N=6-10 for each group; (D) and (E) N=7 for each group; (F) N=6 for each group).

Response: Thanks for the good question. According to Figure 1 A, 10-14 mice for each group (14 for NP-Chow group, 14 for NP-HFD group, 10 for LP-Chow group, 11 for LP-HFD group) were achieved. However, one mouse from NP-HFD group was found to have kidney disease at harvest, and was excluded from all the analysis. Thus, 10-14 mice for each group (14 for NP-Chow group, 13 for NP-HFD group, 10 for LP-Chow group, 11 for LP-HFD group) were monitored for growth performance (Figure 1B).

Among these mice, 3-5 mice per group were random selected for insulin signaling study (5 for NP-Chow group, 4 for NP-HFD group, 3 for LP-Chow group, 4 for LP-HFD group). And then, the other mice (9 for NP-Chow group, 9 for NP-HFD group, 7 for LP-Chow group, 7 for LP-HFD group) were used for blood and tissue collection (Figure 1C). The blood glucose data in the vision 1 was measured by glucose assay kit (Nanjing Jiancheng Bioengineering Institute) with mouse serum. In the revised vision, this figure (Figure 1C) has been updated with the blood glucose levels which were measured by blood Glucose Strip (Beijing Yicheng, Beijing, China) with the tail blood before mouse harvest (line 91-93).

Because the blood glucose levels were measured every 15 min for ITT study, and we usually measure one mouse of the control, and then one of the treatment group 1 min later, thus the total number of mice should be no more than 15. So, we randomly selected 7 mice from each group for ITT study (Figure 1D and E).

Because of well-limitation of Insulin ELISA Kit, we randomly selected 6 samples from each group for serum insulin level analysis. And we believe the result with these samples could represent the changes of the groups.

I feel a bit of lack in the discussion and the conclusion sections, which is probably due to the presentation of the huge amount of data. A good number of individual parameters are mentioned in the discussion, but no broader connections were presented.

Response: Thanks for the valuable suggestion. The discussion has been improved according to the suggestion of the reviewer. The common transcription factors of the Fgf1, Ascl1, Lpl, Cidea and Mup19 genes might mediate the effect of maternal LP diet on the lipids metabolism of offspring, and this will be investigated in further study.

Minor:

there are some typing and/or grammar mistake in the text.

Response: Thanks for the comment. The manuscript has been carefully checked. Typing and grammar mistakes have been improved.

Figure legend 2: “Oil-red O staining and H&E staining Bars equal to 50 μm”, on the pictures bars are mark as 100 μm.

Response: Thanks for the valuable comment. The bars equal to 100 μm. The figure legend has been updated.
